# Trade-offs between individual and ensemble forecasts of an emerging infectious disease

Rachel J. Oidtman [1,2,3✉], Elisa Omodei [2], Moritz U. G. Kraemer [4,5,6], Carlos A. Castañeda-Orjuela [7], Erica Cruz-Rivera[7], Sandra Misnaza-Castrillón[7], Myriam Patricia Cifuentes [8], Luz Emilse Rincon[8], Viviana Cañon[9], Pedro de Alarcon [10], Guido España[1], John H. Huber[1], Sarah C. Hill [4,11], Christopher M. Barker [12], Michael A. Johansson [13], Carrie A. Manore[14], Robert C. Reiner, Jr. [15], Isabel Rodriguez-Barraquer[16], Amir S. Siraj[1], Enrique Frias-Martinez [17], Manuel García-Herranz [2✉] & T. Alex Perkins [1✉]

Probabilistic forecasts play an indispensable role in answering questions about the spread of newly emerged pathogens. However, uncertainties about the epidemiology of emerging pathogens can make it difficult to choose among alternative model structures and assumptions. To assess the potential for uncertainties about emerging pathogens to affect forecasts of their spread, we evaluated the performance 16 forecasting models in the context of the 2015-2016 Zika epidemic in Colombia. Each model featured a different combination of assumptions about human mobility, spatiotemporal variation in transmission potential, and the number of virus introductions. We found that which model assumptions had the most ensemble weight changed through time. We additionally identified a trade-off whereby some individual models outperformed ensemble models early in the epidemic, but on average the ensembles outperformed all individual models. Our results suggest that multiple models spanning uncertainty across alternative assumptions are necessary to obtain robust forecasts for emerging infectious diseases.

[1] Department of Biological Sciences and Eck Institute for Global Health, University of Notre Dame, Notre Dame, IN, USA. [2] UNICEF, New York, NY, USA. [3] Department of Ecology and Evolution, University of Chicago, Chicago, IL, USA. [4] Department of Zoology, University of Oxford, Oxford, UK. [5] Boston Children's Hospital, Boston, MA, USA. [6] Harvard Medical School, Boston, MA, USA. [7] Instituto Nacional de Salud, Bogotá, Colombia. [8] Ministerio de Salud y Protección Social, Bogotá, Colombia. [9] UNICEF, Bogotá, Colombia. [10] LUCA Telefonica Data Unit, Madrid, Spain. [11] Department of Pathobiology and Population Sciences, The Royal Veterinary College, London, UK. [12] Department of Pathology, Microbiology, and Immunology, School of Veterinary Medicince, University of California, Davis, CA, USA. [13] Division of Vector-Borne Diseases, Centers for Disease Control and Prevention, San Juan, Puerto Rico. [14] Information Systems and Modeling (A-1), Los Alamos National Laboratory, Los Alamos, NM, USA. [15] Institute for Health Metrics and Evaluation, University of Washington, Seattle, WA, USA. [16] Department of Medicine, University of California, San Francisco, CA, USA. [17] Telefonica Research, Madrid, Spain. ✉email: rjoidtman@gmail.com; mgarciaherranz@unicef.org; taperkins@nd.edu

Pathogen emergence, or the phenomenon of a novel or established pathogens invading a new host population, has been occurring more frequently in recent decades[1]. In the last 40 years, more than 150 pathogens of humans have been identified as emerging or re-emerging[2,3]. In these situations, host populations are largely susceptible, which can result in dynamics ranging from self-limiting outbreaks, as with Lassa virus[4], to sustained pandemics, as with HIV[5], depending on the pathogen's traits and the context in which it emerges. When emergence does occur, mathematical models can be helpful for anticipating the future course of the pathogen's spread[6–8].

A necessary part of using models to forecast emerging pathogens is making decisions about how to handle the many uncertainties associated with this unfamiliar microbes[8]. Given the biological and ecological diversity of emerging pathogens, there is often considerable uncertainty about various aspects of their natural histories, such as their potential for superspreading[9], the role of human mobility in their spatial spread[10,11], drivers of spatiotemporal variation in their transmission[6,12], and even their modes of transmission[13]. In the case of MERS-CoV, for example, it took years to determine that the primary transmission route was spillover from camels rather than sustained human-to-human transmission[14]. A lack of definitive understanding about such basic aspects of natural history represents a major challenge for forecasting emerging pathogens.

Inevitably, different forecasters make diverse choices about how to address unknown aspects of an emerging pathogen's natural history, as they do for numerous model features. This diversity of approaches has itself been viewed as part of the solution to the problem of model uncertainty, based on the idea that the biases of different models might counteract one another to produce a reliable forecast when viewed from the perspective of an ensemble of models[15]. This idea has support in multi-model efforts to forecast seasonal transmission of endemic pathogens, such as influenza and dengue viruses[16–20], with ensemble forecasts routinely outperforming individual models. These successes with endemic pathogens have motivated multi-model approaches in response to several emerging pathogens, including forecasting challenges for chikungunya[21] and Ebola[22], vaccine trial site selection for Zika[23], and a multi-model decision-making framework for COVID-19[15,24].

Although there has been increased attention to multi-model forecasting of emerging pathogens in the last few years, these initiatives have involved significant effort to coordinate forecasts among multiple modeling groups[25,26]. Coordination across multiple groups has clear potential to add value beyond what any single modeling group can offer alone. At the same time, using multiple models to hedge against uncertainties about a pathogen's natural history could potentially improve forecasts from a single modeling group, too[16,18]. This could, in turn, improve ensemble forecasts based on contributions from multiple modeling groups. An ensemble-based approach by one modeling group that contributes to forecasts of seasonal influenza in the United States demonstrates the success that a single modeling group can achieve with an ensemble-based approach[27] and that such an ensemble can contribute value to an ensemble of forecasts from multiple modeling groups[18]. Similar approaches have not been widely adopted for forecasting emerging pathogens by a single modeling group (although see ref. [28]), despite the heightened uncertainty inherent to emerging pathogens.

Here, we evaluate the potential for an ensemble of models that span uncertainties in pathogen natural history but share a common core structure, to accurately forecast the dynamics of an emerging pathogen. We do so in the context of the 2015–2016 Zika epidemic in Colombia, which was well-characterized epidemiologically (Fig. 1)[29] and involved potentially consequential uncertainties about (i) the role of human mobility in facilitating spread across the country[30], (ii) the relationship between environmental conditions and transmission of this mosquito-borne virus[6,12], and (iii) the number of times the virus was introduced into the country[31]. In this retrospective analysis, we used data assimilation to update 16 distinct models throughout the epidemic period and assessed the forecast performance of all models relative to an equally weighted ensemble model. This allowed us to quantify the contribution of variants of each of the three aforementioned uncertainties to model performance during different phases of the epidemic. In doing so, we sought to not only assess the performance of the ensemble model relative to individual models but also to learn about features of individual models that may be associated with improved forecast accuracy over the course of an epidemic.

## Results

**General forecast performance.** Before any data assimilation had occurred, our 16 models (see Table 1) initially forecasted very low incidence across most departments over the 60-week period of our analysis (Fig. 2 top row, Supplementary Fig. 12). Even so, short-term forecasts over a 4-week horizon were consistent with the still-low observed incidence at that time (Fig. 1 purple, Supplementary Fig. 18). By the time twelve weeks of data had been assimilated into the models, forecasts over the 60-week period of our analysis were considerably higher than the initial forecasts and better aligned with the observed trajectory of the epidemic (Fig. 2 second row, Supplementary Fig. 13). Over those first 12 weeks, model parameters changed modestly (Supplementary Fig. 6) and correlations among parameters began to emerge (Supplementary Figs. 7–10). We observed a more substantial change in the proportion of individual stochastic realizations (where the $n$th stochastic realization is the $n$th "particle" generated from some set of parameters $\vec{\theta}_{t,n}$ at time $t$) resulting in an epidemic, with those particles resulting in no epidemic being filtered out almost entirely by week 12 (Supplementary Fig. 1). Because each particle retained its stochastic realization of past incidence across successive data assimilation periods, stochastic realizations of past incidence were inherited by particles much like parameter values. By week 24, many of the models correctly recognized that they were at or near the epidemic's peak and forecasted a downward trajectory for the remainder of the 60-week period of our analysis (Fig. 2 third row, Supplementary Fig. 27). The particle filtering algorithm replaced nearly half of the original particles by that point (Supplementary Fig. 2), with the new particles consisting of stochastic realizations of past incidence selected through data assimilation and updated every four weeks with forward simulations based on either original or new parameter combinations. As the end of the 60-week period of our analysis was approached, parameter correlations continued to strengthen (Supplementary Figs. 7–10), our estimate of the reporting probability increased (Supplementary Fig. 6), and only around 20% of the original particles remained (Supplementary Fig. 1).

**Model-specific forecast performance.** To quantify the forecast performance of individual models over time, we used logarithmic scoring (hereafter, log scoring) to compare forecasts of cumulative incidence 4 weeks into the future to observed values at departmental and national levels. We assessed log scores once the first case was reported nationally for spatially coupled models (i.e., models with explicit human mobility), and once the first case was reported in each department for nonspatial models (i.e., models with no explicit human mobility). Log scores were generally high for spatially coupled models early in the epidemic, given that observed cases and forecasts were both low at that time (Supplementary

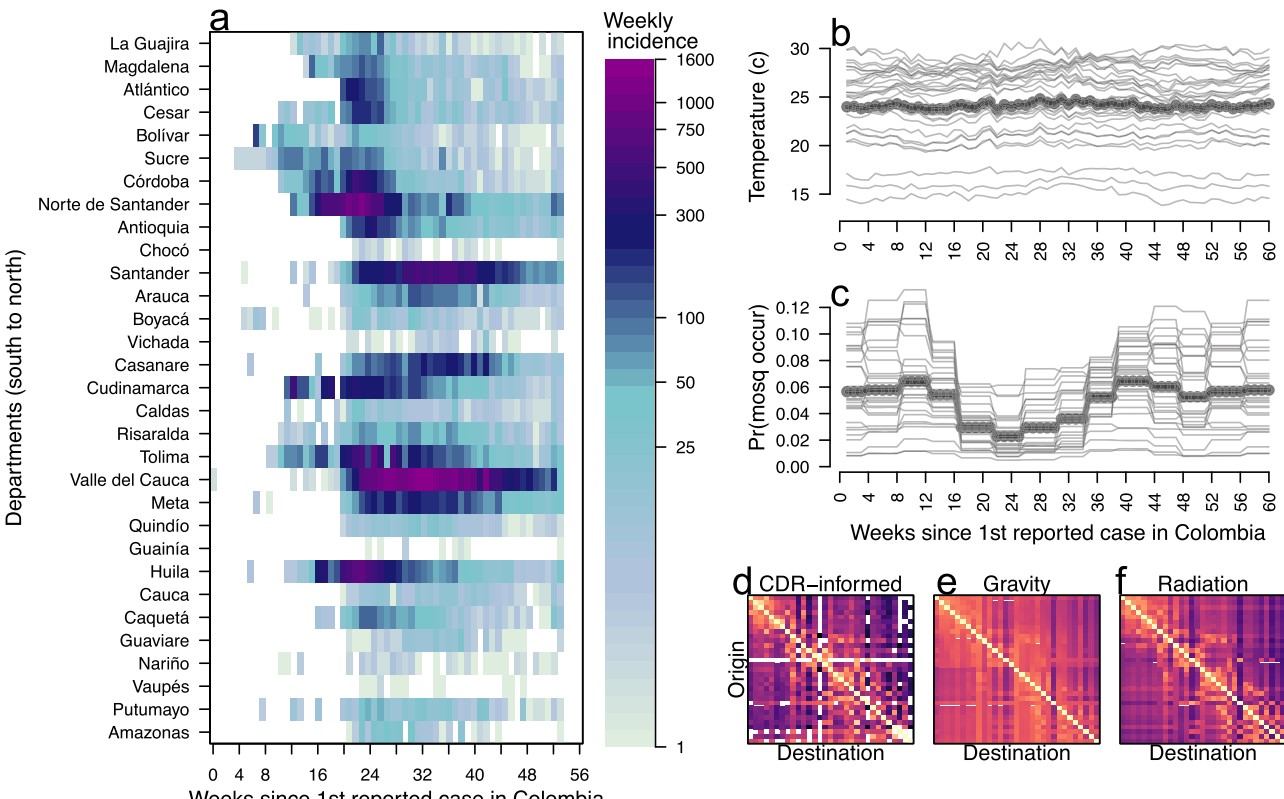

**Fig. 1 Temporal and spatial variation of Zika incidence, temperature, and mosquito occurrence probability in Colombia. a** Weekly Zika incidence from August 9, 2015 to October 1, 2016 with all 31 mainland departments approximately ordered from south to north. **b** Points indicate average temperature data and lines indicate temperature by the department. **c** Points indicate average mosquito occurrence probability and lines indicate mosquito occurrence probability by the department. **d–f** Mobility matrices under three different assumptions of mobility (CDR-informed denotes a mobility matrix derived from call data records), with departments ordered south to north on the y-axis and north–south on the x-axis. Tan indicates high rates of mobility, dark purple indicates low rates of mobility, white indicates no movements.

**Table 1 Different model assumptions regarding the role of human mobility in facilitating pathogen spread across the country, the relationship between environmental conditions and transmission of ZIKV, and the number of times the virus was introduced into Colombia.**

| Human mobility | Transmission potential | Number of ZIKV introductions |
|---|---|---|
| CDR-informed | Fixed $R$[6] | One |
| Gravity model | Dynamic $R$[12] | Two |
| Radiation model | | |
| No human mobility | | |

The suite of 16 models reflected factorial combinations of these three assumptions.

Fig. 18a–c). By week 12, as cases were reported in more departments, the accuracy of forecasts from nonspatial models improved (Supplementary Fig. 18d onward). Forecast performance around the peak of the epidemic differed considerably across models and departments, with forecasts from non-spatial models being somewhat lower than observed incidence and forecasts from spatially coupled models being somewhat higher (Supplementary Fig. 14, Supplementary Fig. 18f–j). Around the peak of the epidemic, forecasts from spatially coupled models generally had higher log scores in departments with lower incidence (e.g., Nariño). Later in the epidemic (weeks 40–56), some models continued to forecast higher incidence than observed in some departments, despite having passed the peak incidence of reported cases (Supplementary

Fig. 16). In particular, models that used the dynamic instead of the static formulation of the reproduction number (i.e., the temporal relationship between $R$ and environmental drivers is dynamic instead of static) were more susceptible to this behavior (note lower log scores in "Rt" versus "R" models in Supplementary Fig. 18k–o), given that their forecasts were sensitive to seasonal changes in temperature and mosquito occurrence.

Next, we used these log scores in an expectation–maximization (EM) optimization algorithm[32] to identify an optimal weighting of retrospective model-specific forecasts into an ensemble forecast (Supplementary Figs. 25–29) in each forecasting period (Supplementary Fig. 17). To learn how model assumptions affected the inclusion of different models into the optimally weighted ensemble for each forecasting period, we summed and then normalized models' ensemble weights across each class of assumption (Fig. 3). Over the course of the epidemic, changes in weighting for the assumptions about human mobility and spatiotemporal variation in transmission, but not about the number of virus introductions into the country, closely followed patterns in the trajectory of the national epidemic. Spatially coupled models had most or all of the weight in the early and late stages of the epidemic, while non-spatial models had most of the weight around the peak of the epidemic (Fig. 3b). Although the non-spatial models somewhat under-predicted incidence in the middle stages of the epidemic, this was often to a lesser extent than the spatially coupled models' over-predictions of incidence (Supplementary Fig. 3). As a result, the EM algorithm achieved a balance between the over- and under-predictions of these different models.

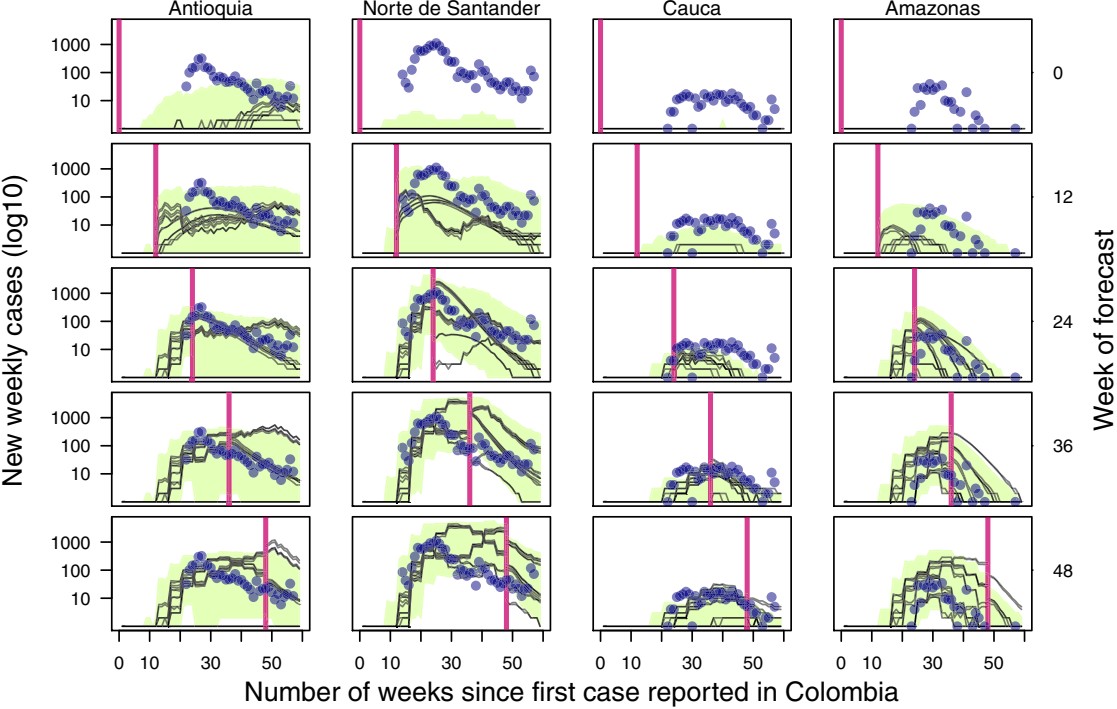

**Fig. 2 Observed incidence (navy points) with the median forecast for 16 models (black lines) with the equally weighted ensemble model (green band) for Antioquia, Norte de Santander, Cauca, and Amazonas at five points throughout the epidemic.** Plotted departments reflect differences in population, epidemic size, and geographic regions of Colombia and are represented by each column. The vertical pink line indicates the point at which the forecast was made (also labeled on the right axis), with data to the left of the line assimilated into the model fit. Forecasts to the right of the vertical line change as more data are assimilated into the model, while the model that fits the left of the vertical line do not change. The green band reflects the 50% credible interval of the equally weighted ensemble model.

The maximum ensemble weight in any forecasting period was 0.802, held by one model with a static $R$, two ZIKV introductions into the country, and CDR-informed human mobility 12 weeks after the first reported Zika case (Supplementary Fig. 17). Combined, the two models with static $R$ and CDR-informed human mobility data had the most instances of a nonzero ensemble weight (Supplementary Fig. 17), occurring in 13 of 15 assimilation periods, with an average weight of 0.18. Around the peak of the epidemic, nonspatial models had the highest ensemble weight, reflecting the accuracy of short-term forecasts in some departments (e.g., Magdalena and Vaupés) and their overall accuracy in nationally aggregated forecasts (Supplementary Fig. 11). Near the end of the epidemic, the ensemble weight for models with a static $R$ (Fig. 3c) increased as their forecasts more closely matched the downturn of incidence later in the epidemic relative to models with dynamic $R$ (Supplementary Fig. 20). This was likely the result of mosquito occurrence probability and temperature becoming more favorable for transmission in many departments later in the epidemic (Supplementary Figs. 21 and 22), causing the dynamic $R$ models to forecast a late resurgence in Zika incidence.

**Target-oriented forecast performance.** Short-term changes in incidence are an important target of infectious disease forecasting, but there are other targets of potentially greater significance to public health decision-making. To explore these, we evaluated the ability of the 16 models—and an evenly weighted ensemble—to forecast three targets at the department level: peak incidence, week of peak incidence, and onset week, which we defined as the week by which ten cases were first reported. We evaluated models based on log scores of these targets. Summing log scores across departments to allow for comparisons across different forecasting periods (Fig. 4), we found

that, on average, the ensemble model outperformed every individual model for all three forecasting targets (indicated by the ensemble model's location on the $y$-axis). Early in the epidemic, spatially coupled models with a static $R$ performed only slightly better (up to 1%) than the equally weighted ensemble (Fig. 4). For the remainder of the epidemic, the equally weighted ensemble model outperformed all individual models (Fig. 4). Such small changes in forecast performance when averaging over space shows that differences in forecast performance across space dominate relative to those across time.

By summing log scores across forecasting periods to allow for comparisons across departments (Fig. 5), we found that some individual models outperformed the ensemble model in forecasting the peak incidence and the week of peak incidence. In departments on the Caribbean Coast that experienced intermediate epidemic sizes (e.g., Antioquia, Sucre, and Atlántico), spatially coupled models with a static $R$ outperformed the ensemble model at forecasting the peak week by about 10% (Fig. 5a). At those same locations, the equally weighted ensemble performed better than or similar to those same models at forecasting peak incidence and onset week (Fig. 5b, c). Over forecasting periods and departments, the nonspatial models consistently had lower average forecast scores than the spatially coupled models (indicated by their location on the $y$-axis in Figs. 4 and 5). This trend appeared because initial forecasts from nonspatial models were not updated until the first case appeared in each department, while initial forecasts from spatially coupled models were updated when the first case appeared in the country.

## Discussion
We assessed the potential for a suite of individual models that span a range of uncertainties, and ensembles of these models, to

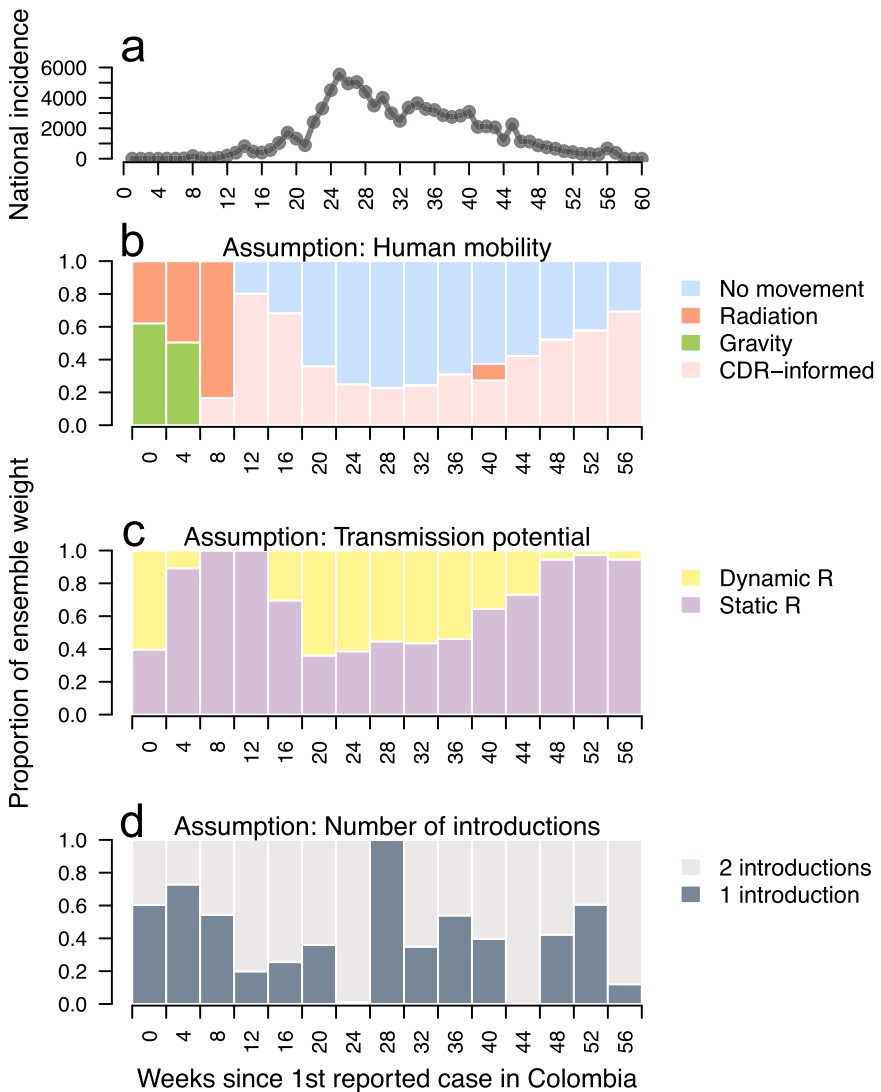

**Fig. 3 Ensemble weight partitioned across assumptions about the role of human mobility in driving transmission, drivers of spatiotemporal variation in R, and the number of ZIKV introductions. a** Weekly Zika incidence aggregated to the national scale. **b–d** Proportion of ensemble weight across assumption type colored by explicit assumption.

accurately forecast the dynamics of an emerging pathogen. Results from the general forecast performance analysis demonstrated that once we began assimilating data into models, forecasts rapidly became more accurate. Models were initialized with a wide range of parameter values[33], with many initial parameter combinations producing unrealistic forecast trajectories, but after only four assimilation periods (12 weeks), nearly 100% of those parameters that produced zero infections were dropped. Similar to other retrospective forecast analyses[16,34], as more data were assimilated into the models over time, the model fits and forecasts generally became more closely aligned with temporal trends in the data. This was because the particle filter allowed model parameters to continually adapt to noisy data[35]. There were still some exceptions where the particle filter could not fully compensate for shortcomings of the transmission model, such as the drastic underestimates of incidence in departments with suboptimal conditions for transmission (e.g., static R model in Risaralda in Supplementary Fig. 20). At the same time, the broader suite of models buffered against shortcomings of any single transmission model.

In the model-specific forecast performance analysis, we identified clear temporal trends related to when models with a

static R versus a dynamic R should be included in an optimally weighted ensemble. In contrast, there were no clear temporal trends in weighting regarding the assumption about the number of times the virus was introduced into the country, potentially reflecting that, even with multiple introductions, the most transmission may have been linked to a single introduction[31]. Models with a dynamic R had higher weights in the ensemble at the peak of the epidemic, while models with a static R had higher weights at the beginning and end of the epidemic. This was likely due to temporal shifts in temperature and mosquito occurrence probabilities dominating forecasts of transmission potential for the models with a dynamic R. For example, in the latter parts of the epidemic when reported cases were declining, mosquito conditions and the temperature became more suitable for transmission in many departments. This caused models with a dynamic R to forecast a resurgence in ZIKV transmission in those departments, while models with a static R forecasted a downturn in incidence that was more similar to the observed dynamics. This finding that susceptible depletion may have been more influential than temporal variation in environmental conditions for the Zika epidemic is consistent with recent findings for SARS-CoV-2[36].

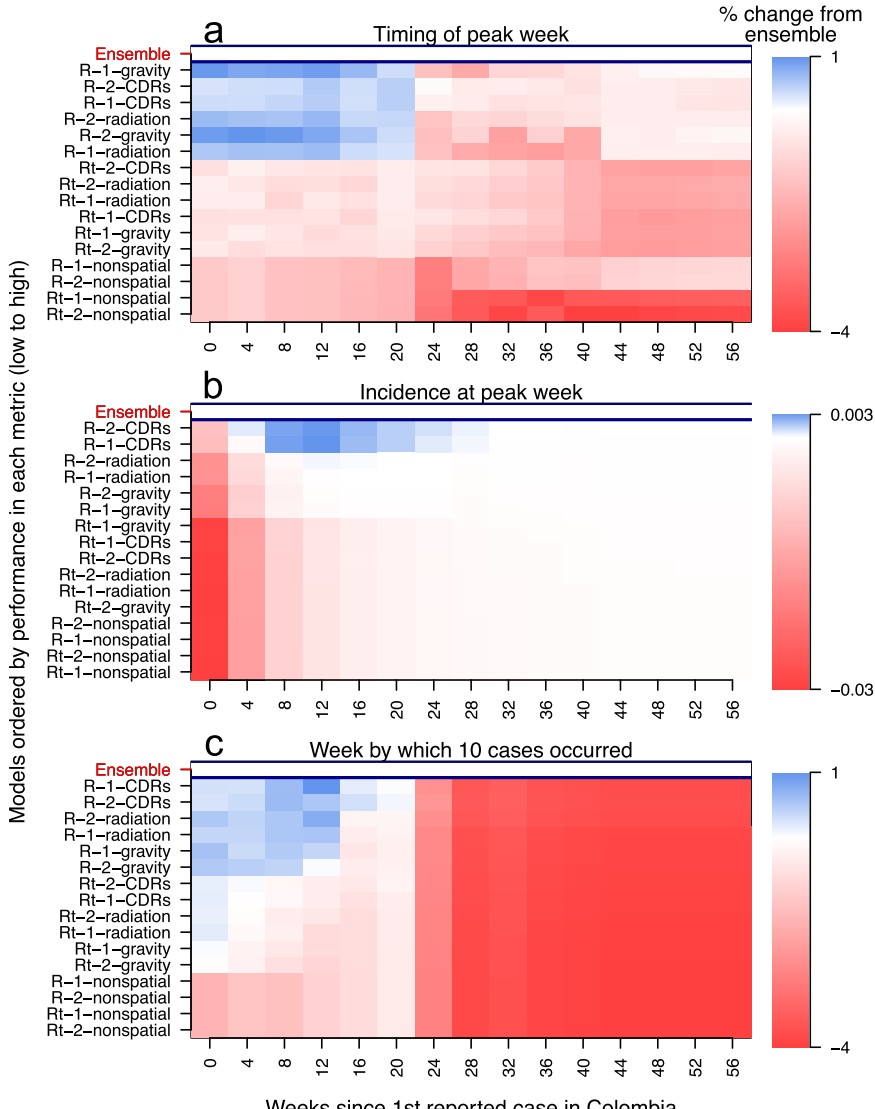

**Fig. 4 Model-specific forecast scores are relative to the equally weighted ensemble model for each assimilation period and forecasting target. a** Timing of peak week (within two weeks). **b** Incidence at peak week. **c** Onset week. Forecast scores are averaged over the department. Models are ordered on the y-axis by the average forecast score for each forecasting target. Model names on the y-axis are abbreviated such that "R" or "Rt" indicates assumption about spatiotemporal variation, "1" or "2" indicates a number of introduction events, and "CDRs", "gravity", "radiation'", or "nonspatial" indicates the human mobility assumption. In the heat plot, blue indicates individual model performed better than the ensemble model in a given department, red indicates individual model performed worse than ensemble model, and white indicates individual model performed roughly the same as the ensemble model.

Through the model-specific forecast performance analysis, we also found that spatially coupled models had higher ensemble weights in the early and late stages of the epidemic, while non-spatial models had higher weights around the peak of the epidemic. The importance of including spatially coupled models in the optimally weighted ensemble early in the epidemic supports the general notion that human mobility may be particularly predictive of pathogen spread early in an epidemic[7,30,37,38]. In part, temporal shifts in weighting around the peak of the epidemic were due to more accurate nationally aggregated forecasts from the non-spatial models. This result was consistent with a previous modeling analysis of the invasion of the chikungunya virus in Colombia, which showed that models fitted independently to subnational time series recreated national-level patterns well when aggregated[39]. A shift in ensemble weights toward non-spatial models around the peak of the epidemic was also due to less accurate department-level forecasts from the spatially

coupled models. At that point in the epidemic, the prevalence was at its highest, which means that we would expect local epidemics to be more endogenously driven and less sensitive to pathogen introductions across departments.

In the target-oriented forecast performance analysis, we found that the equally weighted ensemble generally outperformed individual models, with a few key exceptions. In the months leading up to the peak of the epidemic, spatially coupled models with a static $R$ had slightly, but consistently, higher forecast scores with respect to peak week and onset week. Like the model-specific analysis results, this result illustrates the importance of human mobility in facilitating the spread of an emerging pathogen across a landscape[30]. Individual models outperforming the equally weighted ensemble model in the early phase of the epidemic is not wholly surprising given that non-spatial models were represented equally in that ensemble throughout the epidemic. Non-spatial models may be realistic when locations have self-

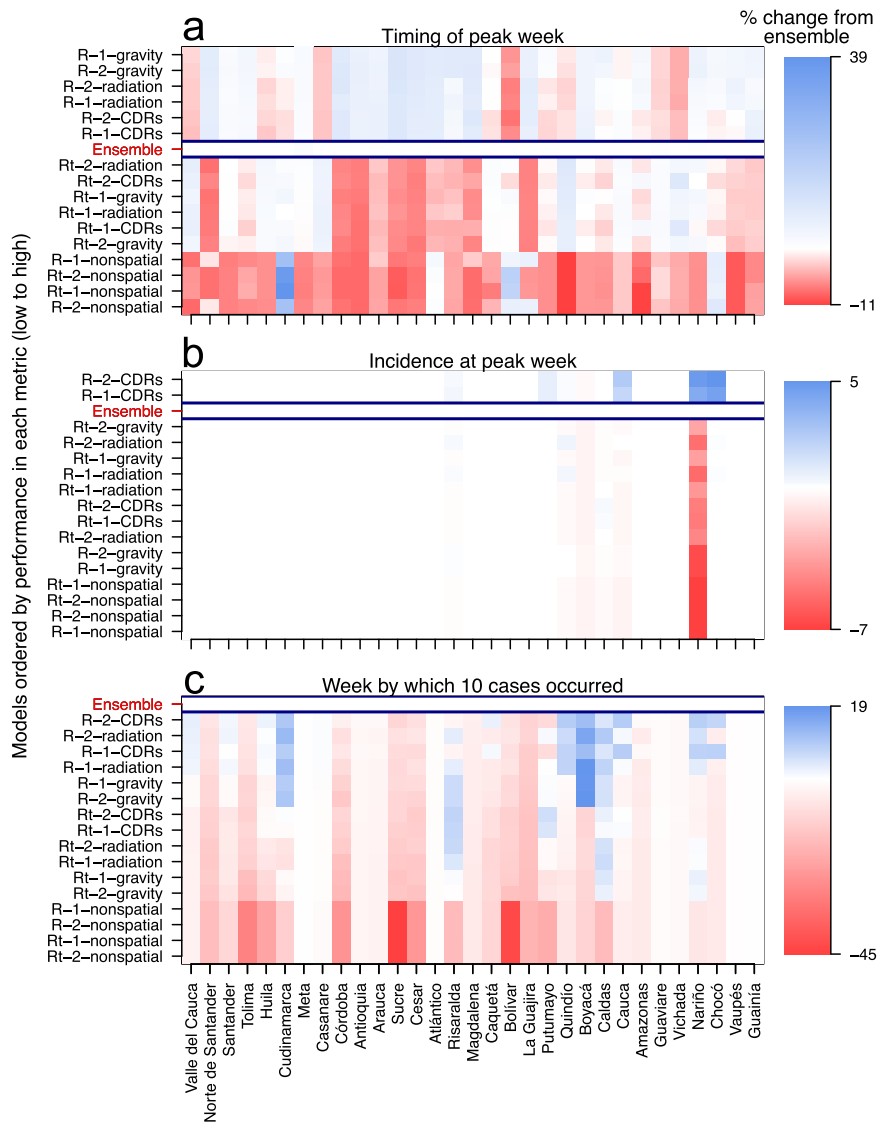

**Fig. 5 Model-specific forecast scores are relative to equally weighted ensemble model for each department and forecasting target. a** Timing of peak week (within 2 weeks). **b** Incidence at peak week. **c** Onset week, or the week by which ten cumulative cases occurred. Forecast scores are averaged over the department. Models are ordered on the y-axis by average forecast score for each forecasting target, with model names abbreviated in the same manner as Fig. 4. Departments are ordered on the x-axis from high to low for overall incidence. In the heat plot, blue indicates individual model performed better than the ensemble model in a given department, red indicates individual model performed worse than ensemble model, and white indicates individual model performed roughly the same as the ensemble model.

sustaining epidemics, but they are not appropriate for capturing early phase growth and its dependence on importations[40]. Another instance when individual models had higher forecast scores than the equally weighted ensemble was with respect to peak week for spatially coupled models with a static $R$ in departments along the Caribbean Coast. Compared to dynamic $R$ models, the static $R$ models more accurately forecasted peak week in these departments (e.g., Magdalena, Cesar, and Sucre), as they did not forecast a late-stage resurgence in transmission. The equal weighting of the dynamic $R$ models in the ensemble, therefore, led to overall lower peak week forecast scores for the ensemble relative to static $R$ models. Still, our results indicating that an equally weighted ensemble mostly outperformed individual models adds to the growing literature highlighting the importance of ensemble methods in epidemiological forecasting[16,17,27,41,42].

We considered both equally and optimally weighted ensembles and found that the equally weighted ensemble had a lower root mean square error than the optimally weighted ensemble (RMSE = 0.640 and 0.705, respectively)—therefore providing slightly more accurate forecasts of the observed data (Supplementary Fig. 23). With the optimally weighted ensemble, which we updated at each data assimilation period, we found that model weights changed over the course of the epidemic Supplementary Fig. 17). Although this is intuitive given the changing nature of an emerging epidemic through time[8], it may be problematic in practice. It is almost as if the ensemble weights require their own forecast. On the one hand, promising new advances in ensemble modeling[27,41]—such as adaptive stacking for seasonal influenza forecasting[43]—are being used to address this issue of identifying optimal, adaptive weights without training to historical data. On

the other hand, is an emerging pathogen context, establishing optimal model weights by way of model fitting and forecast generation is often reliant on available incidence data (rather than historical data) that is highly variable[44], given the delayed nature of data reporting[45]. In this context, our results demonstrate that it is preferable to use an equally weighted ensemble to buffer against uncertainty in optimal ensemble weights. As is also being demonstrated in forecasts of COVID-19, equally weighted ensembles can provide accurate forecasts[26,44,46] and maybe a better reflection of the considerable structural uncertainty inherent to models of emerging pathogen transmission[24].

A few limitations of our study should be noted. First, while an equally weighted ensemble approach allowed us to consider contributions of several alternative model assumptions, there was high uncertainty associated with these forecasts (sometimes spanning orders of magnitude, see Supplementary Fig. 24). Potential end-users of these types of forecasts could consider high levels of uncertainty to be problematic for decision-making[47], though if the uncertainty does not affect the choice of a control measure, then the uncertainty may not be as relevant[48]. In the future, ensemble approaches aimed at increasing precision and reducing uncertainty[27,49] could be used in conjunction with equally weighted ensembles. Second, we considered alternative models across only three assumptions. With ZIKV transmission, there are additional structural uncertainties that could be considered, such as the role of sexual transmission[50]. In real-time applications of our or other Zika forecasting models, it could be worthwhile to explore these types of ZIKV-specific structural uncertainties. Relatedly, the static and dynamic R had minor differences in their formulations, such that the static R also included a socioeconomic index. In future work, it could be interesting to explore if the inclusion of this time-independent variable affected the dynamic R. Third, in this analysis, we did not explicitly consider delays in reporting that likely would have occurred had these forecasts been generated in real time[51]. In that context, temporally aggregating data to a wider interval (e.g., at 2-week intervals rather than 1-week intervals) could potentially help mitigate the effects of reporting delays to some extent. Fourth, we assumed that the reporting probability was constant through time. Although this is a standard assumption[52] given the lack of data to inform a time-varying relationship for this mechanistic element[53], it would be interesting to include and test a reporting dynamics model (e.g., the reporting probability scales with incidence[54]) as an additional component included in our ensemble framework. Fifth, we conducted this analysis at the departmental level instead of this municipality level, which could obfuscate meaningful differences across regions of a single department[29]. In future work, it would be useful to test and assess our forecasting algorithm and outputs at different spatial scales[39].

As the world is reminded of on a daily basis with COVID-19, pathogen emergence is an ongoing phenomenon that will continue to pose threats in the future[55]. A better understanding of an emerging pathogen's natural history could help to reduce pathogen-specific structural uncertainties, but these insights may not always occur in time to inform model development for real-time forecasting[8]. Our results highlight important trade-offs between individual and ensemble models in this context. Specifically, we demonstrated that an equally weighted ensemble forecast was almost always more accurate than individual models. Instances in which individual models were better than the ensemble, or greatly improved the ensemble, also provided insight. For example, incorporating human mobility into models improved forecasts in the early and late phases of an epidemic, which underscores the importance of making aggregated mobility data available early in an epidemic[56]. The range of outcomes resulting from alternative modeling assumptions in model-specific forecasts demonstrates why it will continue to be important to address structural uncertainties in forecasting models in the future.

## Methods

**Data**. We used passive mandatory surveillance data for reported cases of Zika, from the National Surveillance System (Sivigila) at the first administrative level (31 mainland departments) in Colombia. To span the beginning, peak, and tail of the epidemic in Colombia, we focused on the 60-week period between August 9, 2015 and October 1, 2016. We used the version of these data collated by Siraj et al.[29], as well as modeled values of weekly average temperature and estimates of the department-level population from that data set. For some models, we worked with monthly estimates of mosquito occurrence probability (i.e., dynamic R models) obtained from Bogoch et al.[57], and for others, we worked with time-averaged estimates (i.e., static R models) from Kraemer et al.[58].

For models that relied on cell phone data to describe human mobility, we used anonymized and aggregated call detail records (CDRs). Every time a user receives or makes a call, a CDR including the time, date, ID, and the tower (BTS) providing the service is generated. The positions of the BTSs are georeferenced and so the aggregated mobility between towers can be tracked in time. We used this information to derive daily mobility matrices at the municipality level in Colombia from February 2015 to August 2015. Mobility matrices captured the number of individuals that moved in each given day from one municipality to another (i.e., that appeared in BTSs of different municipalities). The change for each day was captured by comparing the last known municipality to the current one. No individual information or records were available.

As these data did not align with the time frame of the epidemic, and to calculate a mobility matrix at a department level, we computed a representative mobility matrix by summing all available CDRs within the municipalities of each department and normalizing them to sum to one relative to the sum of CDRs originating from that department. In five departments (Amazonas, Cudinamarca, Guainía, Vaupés, and Vichada), the proportion of CDRs linking callers within the same department was below 60%. Given that this implied an unrealistically low proportion of time spent within an individual's department of residence, we interpreted those values as idiosyncrasies of the data and not representative of human mobility[59]. Thus, for those five departments, we replaced the proportion of within-department CDRs with the mean proportion of within-department CDRs from all other departments. We then re-normalized the number of CDRs originating from each department in our mobility matrix to sum to one.

**Summary of models**. To produce weekly forecasts of ZIKV transmission across Colombia, we sought to use a computationally efficient model with the flexibility to include relevant epidemiological and ecological mechanisms. We used a previously described semi-mechanistic, discrete-time, and stochastic model[60] that had been adapted and used to model mosquito-borne pathogen transmission[61,62]. Using this model, we were able to account for the extended generation interval of ZIKV using overlapping pathogen generations across up to five weeks of the generation interval distribution of ZIKV[62]. Furthermore, we could specify this model to be either spatially connected or nonspatial—a key assumption that we considered in our analysis.

We considered a suite of 16 models that spanned all combinations of four assumptions about human mobility across Colombia's 31 mainland departments, two assumptions about the relationship between environmental conditions and the reproduction number (R), and two assumptions about how many times the Zika virus was introduced to Colombia (Table 1). Twelve of 16 models allowed for spatial connectivity across departments[60], while four models were nonspatial. There were up to two steps in the transmission process: transmission across departments (for spatially connected models) and local transmission within departments.

Across departments, we simulated the movement of individuals using a spatial connectivity matrix ($\mathbf{H}$), the $d$th column of which corresponds to the proportion of time spent by residents of department $d$ in all departments $\vec{d}$. Using this matrix, we redistributed infections in department $d$ in week $t$ ($I_{d,t}$) across $\vec{d}$ as a multinomial random variable

$$I'_{\vec{d},t} \sim \text{multinomial}(I_{d,t}, \mathbf{H}_{\vec{d},d}), \qquad (1)$$

where the first and second arguments represent the number of trials and the probabilities of the outcomes, respectively. By taking this Lagrangian approach to modeling human mobility, transmission across departments in our model can occur either by infected visitors transmitting to local susceptibles or susceptible visitors becoming infected by local infecteds, but not between infected visitors and susceptible visitors in a transient location. The relative occurrence of these events depends on the prevalence of infection, susceptibility, local transmission potential, and mobility patterns of a given pair of departments.

Within each department, we defined a variable representing the effective number of infections that could have generated new infections in week $t$ ($I''_{d,t}$) as

$$I''_{d,t} = \sum_{j=1}^{5} \omega_j^{GI} I'_{d,t-j}, \qquad (2)$$

where $\omega_j^{GI}$ is the probability that the generation interval is $j$ weeks[63]. The relationship between $I''_{d,t}$ and the expected number of new local infections in week $t+1$ ($I_{d,t+1}$) follows

$$I_{d,t+1} = \beta_{d,t} \frac{I''_{d,t}}{N_d} S_{d,t}, \qquad (3)$$

where $\beta_{d,t}$ is the transmission coefficient, $N_d$ is the total population, and $S_{d,t}$ is the total susceptible population prior to local transmission in week $t$. We accounted for the role of stochasticity in transmission by using the stochastic analog of Eq. (3), such that

$$I_{d,t+1} \sim \text{ negative binomial}\left(\beta_{d,t} \frac{I''_{d,t}}{N_d} S_{d,t}, I''_{d,t}\right) \qquad (4)$$

where the first and second arguments are the mean and dispersion parameters, respectively[60].

To allow for comparison of the model's simulations of infections ($I_{d,t}$) with empirical data on reported cases ($y_{d,t}$), we applied a reporting probability ($\rho$) to simulated infections to obtain simulated cases ($C_{d,t}$), such that $C_{d,t} \sim \text{binomial}(I_{d,t}, \rho)$. Using this, we defined the contribution to the overall log-likelihood of the model and its parameters from a given department $d$ and week $t$ as

$$\ell_{d,t}(\vec{\theta}_t) = \ln\left(\text{negative binomial}(y_{d,t} + 1 \mid \phi, C_{d,t} + 1)\right), \qquad (5)$$

where $\phi$ is a dispersion parameter that we estimated to account for variability in case reporting beyond that captured by $\rho$. Shifting $y_{d,t}$ and $C_{d,t}$ by one in Eq. (5) was intended to safeguard against $\ell_{d,t}$ being undefined in situations where $C_{d,t} = 0$.

*Assumptions about human mobility.* We allowed for spatial coupling across departments in 12 of 16 models. In these models, we informed **H** in three alternative ways: (i) with mobility data extracted from mobile phone CDRs, (ii) with a gravity model, or (iii) with a radiation model (Fig. 1d–f). For the gravity model, we used parameters previously fitted to CDRs from Spain and validated in West Africa[11]. For the radiation model, we calculated human mobility fluxes according to the standard formulation of this model[64], which depends only on the geographic distribution of the population. In four of 16 models, we assumed that departments were spatially uncoupled (Table 1), such that each department was modeled individually with its own set of parameters. In those models, each department's epidemic was seeded independently with its own set of imported infections. Further details about the spatially uncoupled models can be found in the Supplementary Text.

*Assumptions about environmental drivers of transmission.* We parameterized the transmission coefficient, $\beta_{d,t}$, based on a description of the reproduction number, $R_{d,t}$, appropriate to environmental drivers for department $d$ and time $t$. We considered two alternative formulations of $R_{d,t}$ that was informed by data that were available prior to the first reported case of Zika in Colombia. Specifically, both of these alternative formulations used different outputs from previous modeling efforts[6,12], and because of this, they contain slightly different components. Both formulations were defined such that

$$\beta_{d,t} = kR_{d,t} \qquad (6)$$

where $k$ is a scalar that we estimated over the course of the epidemic to account for the unknown magnitude of ZIKV transmission in Colombia. In addition to the summary below, further details about these formulations of $R_{d,t}$ are provided in the Supplementary Methods.

The formulation of $\beta_{d,t}$ that we refer to as "dynamic" is defined at each time $t$ in response to average temperature at that time ($T_{d,t}$) and mosquito occurrence probability at that time ($OP_{d,t}$). This relationship can be expressed generically as

$$\beta_{d,t} = k\bar{R}_{d,t}(T_{d,t}, OP_{d,t}|c, \psi, \alpha, \nu), \qquad (7)$$

where $c$, $\psi$, $\alpha$, and $\nu$ are parameters governing the relationship among $T_{d,t}$, $OP_{d,t}$, and $\bar{R}_{d,t}$. We informed the component of $\bar{R}_{d,t}$ related to mosquito density with monthly estimates of $OP_{d,t}$, which derive from geostatistical modeling of *Aedes aegypti* occurrence records globally[57]. Other components of $\bar{R}_{d,t}$, which include several temperature-dependent transmission parameters, were informed by laboratory estimates[12]. Given that this formulation of $\bar{R}_{d,t}$ was not validated against field data prior to the Zika epidemic in Colombia, we estimated values of $c$, $\psi$, $\alpha$, and $\nu$ over the course of the epidemic.

The formulation of $\beta_{d,t}$ that we refer to as "static" is defined as a time-averaged value that is constant across all times $t$. Temporal variation in $T_{d,t}$ is still accounted for, but its time-varying effect on $R_{d,t}$ is averaged out over all times $\bar{t}$ to result in a temporally constant $R_d$. Mosquito occurrence probability is also incorporated through a temporally constant value ($OP_d$)[58]. The relationship among these

variables can be expressed generically as

$$\beta_{d,t} = k\bar{R}_d(T_{d,\bar{t}}, OP_d, PPP_d), \qquad (8)$$

where $PPP_d$ is purchasing power parity in department $d$ (a feature not included in the dynamic model)[65]. This input is an economic index that was intended to serve as a proxy for spatial variation in conditions that could affect exposure to mosquito biting, such as housing quality or air conditioning use[6]. Given that this formulation of $\bar{R}_d$ was informed by data from outbreaks of Zika and chikungunya prior to the Zika epidemic in Colombia, we did not estimate its underlying parameters over the course of the epidemic in Colombia.

*Assumptions about introduction events.* Although many ZIKV infections were likely imported into Colombia throughout the epidemic, we assumed that ZIKV introductions into either one or two departments drove the establishment of ZIKV in Colombia[31]. Under the two different scenarios, there was either one introduction event into one department or there were two independent introduction events into two randomly drawn departments. For each parameter set, the initial number of imported infections was seeded randomly between one and five in a single week, the timing of which was estimated as a parameter. Following the initial introduction(s), we assumed that ZIKV transmission was driven by a combination of movement of infected people among departments and local transmission within departments, as specified by each model.

**Data assimilation and forecasting.** For each particle, we produced a single forecast to "initialize" the model prior to the first reported case in Colombia. Beginning with the time of the first reported case in Colombia, we then assimilated new data, updated parameter estimates, and generated forecasts every four weeks, consistent with the four-week frequency used by Johansson et al. in an evaluation of dengue forecasts[16]. We specified 20,000 initial parameter sets ($\vec{\theta}_{1,n}$), indexed by $n$, by drawing independent samples from prior distributions of each parameter[66] (see Supplementary Methods). Each parameter set was used to generate a corresponding particle: a stochastic realization of the state variables ($I_{d,1,n}$ and $C_{d,1,n}$). At each assimilation period, we normalized log-likelihoods summed across departments over the preceding four weeks to generate particle weights,

$$\omega(t,n) = \frac{\sum_d \sum_{\tau=t-3}^{t} \ell_{d,\tau}(\vec{\theta}_{t,n})}{\sum_n \sum_d \sum_{\tau=t-3}^{t} \ell_{d,\tau}(\vec{\theta}_{t,n})}. \qquad (9)$$

Proportional to these particle weights ($\omega(t,n)$), we sampled 18,000 sets of corresponding parameters ($\vec{\theta}_t^{\text{resampled}}$) and state variables ($\{I_{d,t}^{\text{resampled}}, C_{d,t}^{\text{resampled}}\}$) from time $t$ with replacement and used them at the next data assimilation step four weeks later, where boldface indicates a set of parameters or state variables, respectively, overall $n$. In doing so, information including the initial prior assumptions ($\vec{\theta}_{1,n}$) and the likelihoods at each four-week period was assimilated into the model sequentially over time. Given that particle filtering algorithms are susceptible to particle drift—or the convergence of particles onto very few states through iterative resampling[33]—we also generated 2,000 new parameter sets at each data assimilation step. To do so, we drew random samples of model parameters from a multivariate normal distribution with parameter means and covariances fitted to the resampled 18,000 parameter sets ($\vec{\theta}_t^{\text{resampled}}$). Whereas the 18,000 resampled parameter sets already included simulated values of state variables $I_{d,t,n}$ and $C_{d,t,n}$ through time $t$, the 2000 new parameter sets did not and so we informed initial conditions of $I_{d,t}^{\text{new}}$ with draws from $I_{d,t}^{\text{resampled}}$ for those parameter sets at the time they were created. Together, the 18,000 resampled parameter sets ($\vec{\theta}_t^{\text{resampled}}$) and the 2000 new parameter sets ($\vec{\theta}_t^{\text{new}}$) constituted the set of parameter sets used as input for the next data assimilation step ($\vec{\theta}_{t+4} = \{\vec{\theta}_t^{\text{resampled}}, \vec{\theta}_t^{\text{new}}\}$). We also used this new set of parameters as the basis for forecasts made at time $t$, which simply involved simulating forward a single realization of the model for each parameter set.

**Evaluating forecast performance.** At each of the 15-time points at which we performed data assimilation through the 60-week forecasting period, we created an ensemble forecast that evenly weighted contributions from each of the 16 models[46]. To populate this ensemble, we specified 20,000 total samples, with 1250 samples from each model. We assessed the model-specific performance of individual and ensemble forecasts using log scores, which are forecast scoring rules that assess both the precision and accuracy of forecasts[67]. For a specific forecasting target, $z$, and model, $m$, the log score is defined as $\log f_m(z^*|\mathbf{x})$, where $f_m(z|\mathbf{x})$ is the predicted density conditioned on the data, $\mathbf{x}$, and $z^*$ is the empirical value of the target $Z$[16].

We computed log scores for departmental and national incidence over each four-week assimilation period. Following[17], we used an EM algorithm to generate ensemble weights for each model in each assimilation period. For each model, we computed 32 log scores (i.e., one for each department and one nationally). To compute the ensemble weight for a given model feature, such as the static $R$ assumption, we summed the weights of all models with that feature.

We assessed target-oriented forecast performance using log scores for three forecasting targets: timing of peak week (within two weeks), incidence at peak week, and onset week, which we defined as the week by which ten cumulative cases

occurred. These choices were motivated by forecasting assessments for influenza and dengue[16–18,68] and deemed applicable to public health objectives for forecasting an emerging pathogen such as Zika. For peak week and onset week, we used modified log scores[18], such that predictions within two weeks of the correct week were considered accurate. We evaluated a total of 7680 log scores, reflecting three targets for each of 16 models in each of 31 departments plus at the national level and at each of 15-time points at which data assimilation occurred.

As log scores only yield a relative measure of model performance, we used forecasting scores[18] as a way to retrospectively compare forecast performance for different forecasting targets. Whereas log scores are preferable for comparing performance across models on the same data, forecasting scores are preferable for comparing forecast performance across data composed of different locations and times. A forecasting score is defined simply as the exponential of the average log score, where the latter reflects an average over one or more indices, such as models, time points, targets, or locations.

**Reporting summary**. Further information on research design is available in the Nature Research Reporting Summary linked to this article.

## Data availability
The mobile phone data set used in this study is proprietary and subject to strict privacy regulations. Access to this data set was granted after reaching a non-disclosure agreement with the proprietor, who anonymized and aggregated the original data before giving access to the authors. Access to the dataset is controlled and restricted under strict security and privacy measures due to the company's policy towards preserving customer's data privacy (in accordance with existing data protection regulations) as well as protecting business secrecy. The data could be available on request after negotiation of a non-disclosure agreement. The response to any request shall be provided within the next 15 business days. The contact person is Pedro A. de Alarcón (pedroantoniode.alarconsanchez@telefonica.com). The epidemiological, meteorological, and demographic data are publicly available from Siraj et al.[29] (Dryad repository: https://doi.org/10.5061/dryad.83nj1) and additionally available on GitHub (https://github.com/roidtman/eid_ensemble_forecasting).

## Code availability
The code used to fit models, produce forecasts, analyze forecast outputs, and produce figures are available on GitHub (https://github.com/roidtman/eid_ensemble_forecasting) and Zenodo (https://doi.org/10.5281/zenodo.5176776).

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

## Acknowledgements

The authors would like to thank Clara Palau Montava for help with managing the early stages of this project and Chris Fabian, Evan Wheeler, and Vedran Sekara for comments, suggestions, and support throughout the duration of this project. The authors would like to thank Sebastian Baña for technical support in running models on the Databricks computing platform. The authors would additionally like to thank the UNICEF-Colombia Representative, Aida Oliver Arostegui, INS Director, Martha Lucia Ospina Martinez, and the past and present Ministers of the Colombia Ministry of Health, Juan Pablo Uribe Restrepo and Fernado Ruiz Gomez. Lastly, the authors would like to thank two anonymous reviewers for their constructive comments and useful suggestions.

R.J.O. acknowledges support from an Eck Institute for Global Health Fellowship, GLOBES grant, Arthur J. Schmitt Fellowship, and the UNICEF Office of Innovation. M.U.G.K. is supported by The Branco Weiss Fellowship—Society in Science, administered by the ETH Zurich and acknowledges funding from the Oxford Martin School and the European Union Horizon 2020 project MOOD (#874850). J.H.H. acknowledges funding from a National Science Foundation Graduate Research Fellowship and a Richard and Peggy Notebaert Premier Fellowship. S.C.H. is supported by the Wellcome Trust (220414/Z/20/Z). This research was funded in whole, or in part, by the Wellcome Trust [Grant no. 220414/Z/20/Z]. For the purpose of open access, the author has applied a CC BY public copyright license to any Author Accepted paper version arising from this submission. CMB, MAJ, CAM, RCR Jr., IR-B, ASS, and TAP were supported by a RAPID grant from the National Science Foundation (DEB 1641130).

## Author contributions

R.J.O., E.O., M.U.G.K., C.M.B., M.A.J., C.A.M., R.C.R., I.R.-B., M.G.-H., and T.A.P. conceptualized the study; R.J.O., E.O., M.U.G.K., C.A.C.-O., E.C.-R., A.M.-C., P.C., L.E.R., V.C., P.A., G.E., J.H.H., S.C.H., A.S.S., E.F.-M., and M.G.-H. provided and/or processed data; R.J.O., E.O., M.U.G.K., C.A.-O., E.C.-R., S.M.-C., P.C., L.E.R., V.C., P.A., E.F.-M., M.G.-H., and T.A.P. participated in biweekly meetings to provide feedback on research; R.J.O., E.O., M.U.G.K., M.G.-H., and T.A.P. developed the model and wrote the first draft of the paper; R.J.O., E.O., M.U.G.K., J.H.H., and S.C.H. analyzed the data; E.O., M.G.-H., and T.A.P. supervised the research; all authors reviewed the paper.

## Competing interests

The authors declare no competing interests.
