## [Peer Review File · Nature Communications]

Trade-offs between individual and ensemble forecasts of an emerging infectious diseaseREVIEWER COMMENTS

Reviewer #1 (Remarks to the Author):

Nat Comm Review Oidtman

I found this to be a very nice and comprehensive analysis. I have a few concerns that I have outlined below.

1. It appears that the authors are attempting to do two things in this manuscript -- use the ensemble to infer mechanism using the ensemble weights and make accurate forecasts. These two messages seem to fight with each other in the discussion, where the authors seem to advocate strongly for equal weighting. If I am reading correctly, I think the message could be restructured to highlight that these two objectives are not necessarily in conflict. However, as stated now, the argument that the equal weighted ensemble generates better forecasts (L189-190) seems to cast doubt on the mechanistic inference from the ensemble weights.

I found this apparent conflict between the two streams of results distracting and I would encourage the authors to give some thought to reorganizing the presentation to address these two applications explicitly. The subheadings "Model Specific Forecast Performance" and "Target-Oriented Forecast Performance" seem like they are naturally oriented to these two applications, but are motivated in terms of the methods used rather than the application. If it were me (so taken with the necessary grain of salt) I would re-title these sections to reflect the application and interpretation of results (the former discusses inference about model structure, the latter about forecast accuracy).

L 196: What methods are you contrasting this to? I naturally think of comparing forecasts to observation to generate ensemble weights. If the data in emerging infections are too poor to fit models to, then I don't think the ensemble method is the issue. I don't see how equal weighting overcomes the problem of weak data.

L 206: In decision-making applications, the explicit forecast may not necessarily be the most relevant issue. Rather it is the corresponding decision that is recommended across the ensemble of models -- in principle the forecasts could all be biased by an unknown constant but still rank interventions correctly. See Li et al 2017 PNAS "Essential information: Uncertainty and optimal control of Ebola outbreaks" for an example of this. Note, of course that interventions that depend on absolute value of the forecast (e.g. hospital capacity) are more complicated.

L 206: it would be good to see a summary of the results on the magnitude of the uncertainty bounds for these different methods.

L265: "regardless of infection status". Does the model allow movement of anyone other than infected individuals? Is there movement of susceptibles here? It seems from the prior line that only infected individuals are moving, unless I missed something.

L276: An interesting extension of this work would be to consider that the reporting rate is not constant. In general, that's a very hard thing to do, which is why we always assume it is constant. But in the context of your "model specific forecast performance" you could include alternated reporting dynamics models (e.g. reporting scales with incidence). This is one of those gnarly mechanistic elements (like movement structure, variable R, or initial conditions) that we often make strict assumptions about for the sake of convenience. But there doesn't appear to be any reason it shouldn't be considered within the ensemble framework.

L330: "particle values" - this may be a discipline specific difference, but in much of the statistics literature on particle filter, particles only refer to the state values, and not the associated parameters. Classically, particle filter is used to approximate the likelihood for a specific set of parameters. Doing that over many parameters allows one to find an approximate maximum likelihood estimate (with similar analog in Bayes). Here it sounds like you are combining Bayes

(drawing from the prior) with particle filter (evaluating the likelihood at that prior draw). I am not as familiar with how these terms are used in the forecasting literature, but readers with my background may find this definition and discussion of "drawing" particles from a prior confusing. It would help to at least define the vector θ and its elements and clarify what, if anything, changes over t .

L364-5: "within two weeks" this should be clarified in the results text, perhaps in the figure legend. Without stating this, a reader that only goes through the main text will assume Fig 4/5 is measuring timing of the peak explicitly.

Fig S3/11/23 -- these are complicated figures, so any presentation will require a trade off. But since these are presentations of time series, I would consider a presentation that connects points over time. I find that color is the dominant pattern that I see, which connects points across space within a given time. So to me, rather than being able to see whether the set of orange-blue-yellow-red-green points line up on the 1:1 line (a good prediction), I see that the estimates at time 2 (red) in a given location are all over the place. One option would be to use color for location and symbol for time. An alternative would just be to plot lines -- thus projections for a location are connected as a time series, then use different colors for each location (this may get messy with trying to represent variability as multiple lines, but possible with transparency?)

Fig S4: since this is about the difference between prior and posterior you could just present this as the absolute difference. That would still illustrate the settings for which the estimate moved from the prior. Right now, the diverging color scale makes a tiny positive move look as significant as a large negative move. I would also recommend standardizing this given the differences in magnitude for the parameters -- e.g. divide the change by the variance of the prior, or something similar.

Reviewer #2 (Remarks to the Author):

The manuscript by Dr. Oidtman and colleagues describes an ensemble-modeling framework for the forecast of epidemiological dynamics associated with an emerging infectious disease. Specific reference is made to the Zika epidemic that hit Colombia in 2015-2016. The authors use a set of models whose features span three main axes: spatial structure (coupled vs. uncoupled, in the former case according to various hypotheses), temporal variability of the force of infection (dynamics vs. constant), and number of introduction events (one vs. two). Individual models are calibrated against surveillance data using a particle-filtering algorithm, which allows progressive assimilation of epidemiological information during the course of an outbreak in the form of updated estimates of the model parameters. In addition to the individual models' projections, also evaluated are the performances of ensembles obtained with either equal or suitably optimized weights. Performance is assessed in terms of the ability of each model (individual or ensemble) to forecast in space and time disease incidence or specific quantities of epidemiological interest. The analysis of different families of models and ensembles thereof allows the identification of possible trade-offs between the forecasting power of certain individuals models for specific epidemiological patterns vs. the overall potential of ensemble models averaged over the whole course of the epidemic. The take-home message is that ensembles of models accounting for different sources of uncertainty may be key to robust forecasts in the context of emerging infectious diseases. Although not completely a new one, this undoubtedly is an important message---also one that does ring a bell at a point in time when we all are facing the daily uncertainties associated with a pandemic caused by an emerging pathogen. The manuscript is quite well written and suitably organized; it is also supported by an extensive set of supplementary results.

All this being said, I have some technical comments that I would like to see addressed by the authors:

- The basic structure of the models is not completely clear, at least to me. From equation (4), one gathers that $I_{d,t}$ depends on $I'_{d,t}$, which, from equation (2), is defined as a function of $I'_{d,t-j}$

($j=1\dots 5$), which in turn is defined in equation (1) as a function on I_d, t . The lack of an epidemiological description of the underlying process does not help to solve this seeming circularity in the definition of disease incidence. I would suggest clarifying this methodological aspect that is common to all models. Relatedly, why did I''_d, t also correspond to the dispersion parameter of the NB distribution in equation (4)?

- Concerning human mobility, I understand that there might be some issues related to confidentiality, but the description of how mobile phone traces were used to inform spatial connectivity is quite obscure. For instance, what kind of information was stored in the CDRs? How was matrix H estimated? From line 246, it seems that the strength of pairwise spatial connections was assumed to be proportional to the number of calls between individuals living in (or calling from?) two (different) spatial units: is it so, and what is the rationale? Was it necessary to run a home-detection algorithm or was this piece of information readily available in the CDRs? In general, a more detailed (yet, obviously, fully privacy- and confidentiality-compliant) description of the CDRs and their use seems to be warranted, here.

- As for the environmental drivers of transmission, I wonder whether some form of "hybridization" between the dynamic and static approaches has been attempted. The question arises because the static formulation includes, in addition to time-averaged proxies of temperature and vector abundance, a socioeconomic index that does not appear in the dynamic formulation. What is the reason for this difference? From the Methods section in the main text, I imagined it came from previous modeling efforts; from the Supplementary Information, I gathered that different "building bricks" were put together by the authors, resulting in the two different formulations. I believe that the reader should be given some more background information to fully appreciate the modeling choices made by the authors in this respect.

- The different model structures are characterized by different sets of parameters in need of calibration, each characterized by a specific cardinality. I wonder whether this has been (or can be) in any way used in the definition of the weights in ensemble forecasts.

- I am a bit skeptical about producing forecasts of any sort before performing data assimilation (e.g. Figure 2). What is the merit in that, besides of course setting a baseline for the progressively increasing ability of the models to anticipate the epidemiological dynamics? Evidently, the choice of the initial prior distributions is not without consequences from a quantitative point of view, as the authors show that up to 20% of the originally sampled particles (hence, before the first data assimilation step) may remain until the end of the simulation period.

- I would have expected a more detailed analysis of the performances of the equally- vs. optimally-weighted models. The authors state (line 190) that they generally found better performances associated with the former, but little quantitative evidence seems to be reported in the main text to support their observation. In fact, the optimally-weighted model is only used in Figure 3, where its overall forecast abilities cannot even be fully appreciated. By contrast, Figures 2, 4, and 5 only report results for the single models and the equally weighted ensemble. I think this comparison deserves a more quantitative assessment, also given its implications in the context of emerging pathogens, for which a "simpler" (and perhaps less data-greedy) equally-weighted approach sounds clearly appealing.

Minor comments

- line 90: why "retrospective"? This term and "real-time" seem to be used in opposition to describe target-oriented vs. incidence forecasts. However, the use of the two terms does not appear fully consistent. I would suggest sticking to consistent wording to help the reader navigate the different numerical exercises

- Figure 4: those % changes are very small: can you comment on that?

- line 322: in the Methods section, which I read before the Results, the term "particle" is still undefined (of course the context becomes clear around line 330). I would suggest replacing it here with something different

- line 366: I know this will sound nitpicky, but $7,680 = 16 * 366 * 15 * (31 + 1)$, i.e. to get to the

indicated total you need to include also the log scores at the national levels

Comments are in black and responses are in blue. Excerpts from the manuscript are italicized.

Reviewer #1 (Remarks to the Author):

Nat Comm Review Oidtmán

I found this to be a very nice and comprehensive analysis. I have a few concerns that I have outlined below.

Comment 1.1

1. It appears that the authors are attempting to do two things in this manuscript -- use the ensemble to infer mechanism using the ensemble weights and make accurate forecasts. These two messages seem to fight with each other in the discussion, where the authors seem to advocate strongly for equal weighting. If I am reading correctly, I think the message could be restructured to highlight that these two objectives are not necessarily in conflict. However, as stated now, the argument that the equal weighted ensemble generates better forecasts (L189-190) seems to cast doubt on the mechanistic inference from the ensemble weights.

I found this apparent conflict between the two streams of results distracting and I would encourage the authors to give some thought to reorganizing the presentation to address these two applications explicitly. The subheadings "Model Specific Forecast Performance" and "Target-Oriented Forecast Performance" seem like they are naturally oriented to these two applications, but are motivated in terms of the methods used rather than the application. If it were me (so taken with the necessary grain of salt) I would re-title these sections to reflect the application and interpretation of results (the former discusses inference about model structure, the latter about forecast accuracy).

Response 1.1

The goal of this study was to make accurate forecasts. We were specifically interested in assessing the potential for a suite of models that spanned uncertainties about the pathogen's natural history to accurately forecast the dynamics of an emerging pathogen. Ultimately, inferring mechanisms that contribute to the most accurate forecasts was a necessary stepping stone. At the same time, our inferences revealed intuitive results, such as the addition of human movement in models improving forecasts early on in the epidemic, which we thought it relevant to highlight as a useful takeaway for readers.

We appreciate the reviewer's identification of tension around this issue so that we can better clarify our purpose in this regard. We did consider changing the subheadings, but ultimately decided to keep the headings the same as we think they provided more clarity than other alternatives. We did add a sentence to the beginning of the Discussion to reiterate the goal of our study (lines 181-182), and we tried to adjust wording throughout to highlight our goal of accurate forecasts (e.g., line 51, lines 121-123). We also revised the text in the paragraph comparing optimally weighted and equally weighted models to highlight that *both* types of ensembles produced accurate forecasts (with similar root mean square errors), but that the equally weighted ensemble might be more robust to anomalies in data reporting:

We considered both equally- and optimally weighted ensembles and found that the equally-weighted ensemble had a lower root mean square error than the optimally weighted ensemble (RMSE=0.640 and 0.705, respectively)—therefore providing slightly more accurate forecasts of the observed data (Fig. S23).

Lastly, we aren't necessarily trying to suggest that one ensemble is inherently better or worse, but that the equally weighted ensemble, given that it doesn't require assessing forecasts relative to (potentially anomalous) data through time, is the "safer bet." We hope that this message is clearer now.

Comment 1.2

L 196: What methods are you contrasting this to? I naturally think of comparing forecasts to observation to generate ensemble weights. If the data in emerging infections are too poor to fit models to, then I don't think the ensemble method is the issue. I don't see how equal weighting overcomes the problem of weak data.

Response 1.2

We updated this sentence to clarify and added some additional citations to the paragraph as a whole.

On the other hand, in an emerging pathogen context, establishing optimal model weights by way of model fitting and forecast generation is often reliant on available incidence data (rather than historical data) that is highly variable, given the delayed nature of data reporting (Perkins et al. 2020).

We are not necessarily trying to argue that the data is “weak,” but rather that it is highly variable with potential delays and anomalies present due to data reporting issues. In this regard, the equally weighted model may be more robust to outlier forecasts (from those models that were sensitive to data anomalies) compared to the optimally weighted forecast (which would potentially weight the outlier forecast more heavily given the correspondence with the data anomaly).

Comment 1.3

L 206: In decision-making applications, the explicit forecast may not necessarily be the most relevant issue. Rather it is the corresponding decision that is recommended across the ensemble of models -- in principle the forecasts could all be biased by an unknown constant but still rank interventions correctly. See Li et al 2017 PNAS "Essential information: Uncertainty and optimal control of Ebola outbreaks" for an example of this. Note, of course that interventions that depend on absolute value of the forecast (e.g. hospital capacity) are more complicated.

Response 1.3

This is an important point that we had missed before. We updated the text on lines 276-279 based on this suggestion:

Potential end-users of these types of forecasts could consider high levels of uncertainty to be problematic for decision-making (Bodner et al., 2020), though if the uncertainty does not affect the choice of action, then the uncertainty may not be as relevant (Li et al., 2017).

Comment 1.4

L 206: it would be good to see a summary of the results on the magnitude of the uncertainty bounds for these different methods.

Response 1.4

We added Figure S24 to the supplement as an example of how the magnitude of uncertainty changes through time (referenced on lines 276). This figure corresponds to the example given in Figure S23 comparing the equally and optimally weighted ensemble.

Magnitude of the 50% uncertainty bounds (as shown in Fig. S23) for 1-, 2-, 3-, and 4- week ahead forecasts in five different data assimilation periods for the EM-weighted and equally weighted ensemble models for Amazonas, Antioquia, Cauca, and Norte de Santander. The four points per data assimilation period represent the 1-, 2-, 3-, and 4- week ahead forecasts for each of the four departments denoted by color. Smoothed loess lines are shown to demonstrate how the magnitude of uncertainty changes through time for each department.

Comment 1.5

L265: "regardless of infection status". Does the model allow movement of anyone other than infected individuals? Is there movement of susceptibles here? It seems from the prior line that only infected individuals are moving, unless I missed something.

Response 1.5

We apologize for the confusion around this issue. In short, our approach to modeling human mobility and transmission related thereto is a standard Lagrangian approach (i.e., commuting patterns whereby people do not relocate, as opposed to an Eulerian approach involving fluxes and redistribution of population). In addition to some minor wording changes to reduce the references to infection status, we added the following statement just after the presentation of the equation for transmission across departments to clarify this:

By taking this Lagrangian approach to modeling human mobility, transmission across departments can occur either by infected visitors transmitting to local susceptibles or susceptible visitors becoming infected by local infecteds. The relative occurrence of these events depends on the prevalence of infection, susceptibility, local transmission potential, and mobility patterns of a given pair of departments.

Comment 1.6

L276: An interesting extension of this work would be to consider that the reporting rate is not constant. In general, that's a very hard thing to do, which is why we always assume it is constant. But in the context of your "model specific forecast performance" you could include alternated reporting dynamics models (e.g. reporting scales with incidence). This is one of those gnarly mechanistic elements (like movement structure, variable R, or initial conditions) that we often make strict assumptions about for the sake of convenience. But there doesn't appear to be any reason it shouldn't be considered within the ensemble framework.

Response 1.6

This is an interesting idea. Although we allow for the constant reporting probability to be updated as new data is assimilated into the model, including a time-varying component to the reporting probability is something we had not previously considered. We included this as a potential limitation to our work and noted in the Discussion that it would be an interesting research avenue to explore in future work on lines 293-298:

Fourth, we assumed that the reporting probability was constant through time. Although this is a standard assumption [52] given the lack of data to inform a time-varying relationship for this mechanistic element [53], it would be interesting to include and test a reporting dynamics model (e.g., the reporting probability scales with incidence [54]) as an additional component include in our ensemble framework.

Comment 1.7

L330: "particle values" - this may be a discipline specific difference, but in much of the statistics literature on particle filter, particles only refer to the state values, and not the associated parameters. Classically, particle filter is used to approximate the likelihood for a specific set of parameters. Doing that over many parameters allows one to find an approximate maximum likelihood estimate (with similar analog in Bayes). Here it sounds like you are combining Bayes (drawing from the prior) with particle filter (evaluating the likelihood at that prior draw). I am not as familiar with how these terms are used in the forecasting literature, but readers with my background may find this definition and discussion of "drawing" particles from a prior confusing. It would help to at least define the vector theta and its elements and clarify what, if anything, changes over t.

Response 1.7

This is an important clarification that we were not previously making. We added a definition on line 77-81 (when we first introduce particles) to clarify what we mean by "particle":

We observed a more substantial change in the proportion of individual stochastic realizations (where the n^{th} stochastic realization is the n^{th} "particle" generated from some set of parameters $\vec{\theta}_{t,n}$ at time t) resulting in an epidemic, with those particles resulting in no epidemic being filtered out almost entirely by week 12.

Throughout the "Data assimilation and forecasting" section, we made significant updates to the terminology to help with this clarification. We specifically denoted the difference between a parameter set ($\vec{\theta}_{t,n}$) and a particle ($\{I_{d,t,n}, C_{d,t,n}\}$), where a particle is the n^{th} set of state variables associated with the n^{th} parameter set. Throughout, we stopped using the term "particle" as loosely as we did previously.

Comment 1.8

L364-5: "within two weeks" this should be clarified in the results text, perhaps in the figure legend. Without stating this, a reader that only goes through the main text will assume Fig 4/5 is measuring timing of the peak explicitly.

Response 1.8

We added this to Figure 4 and 5 legends.

Comment 1.9

Fig S3/11/23 -- these are complicated figures, so any presentation will require a trade off. But since these are presentations of time series, I would consider a presentation that connects points over time. I find that color is the dominant pattern that I see, which connects points across space within a given time. So to me, rather than being able to see whether the set of orange-blue-yellow-red-green points line up on the 1:1 line (a good prediction), I see that the estimates at time 2 (red) in a given location are all over the place. One option would be to use color for location and symbol for time. An alternative would just be to plot lines -- thus projections for a location are connected as a time series, then use different colors for each location (this may get messy with trying to represent variability as multiple lines, but possible with transparency?)

Response 1.9

We looked into this and tested a few alternative options. Ultimately, we decided to keep the original figures. The goal of this figure is to compare forecasts for different points in the epidemic for the different models with several examples across departments. We actually want to highlight that (for example)

estimates at time 2 (red) vary depending on location and model, as this helps to support that certain models do better at certain points in time. When we alternatively produced these figures with a different color for location and different symbol for time, we thought it was too difficult to see how forecasts differed at different points in the epidemic and instead the focus was on how forecasts differed at different locations.

At the same time, given the complexity of these types of figures, we included the individual-model and ensemble model forecasts for every location for these time points in the supplement to provide a different way to visualize forecast accuracy over time.

Comment 1.10

Fig S4: since this is about the difference between prior and posterior you could just present this as the absolute difference. That would still illustrate the settings for which the estimate moved from the prior. Right now, the diverging color scale makes a tiny positive move look as significant as a large negative move. I would also recommend standardizing this given the differences in magnitude for the parameters - e.g. divide the change by the variance of the prior, or something similar.

Response 1.10

This is a very good suggestion. We adjusted Fig. S4 to standardize across the differences in magnitude across parameters by dividing the change by the mean of the prior. This adjustment changed the range of the color bar to -0.5 to 1.6 from -12 to 0.19 and ultimately makes the figure much more accessible. We ultimately changed the color scheme of this figure (and Fig 4-5) in light of a suggestion from a colorblind colleague.

Relative difference between the prior estimates of the parameters and the posterior estimates at the final time point in the epidemic. Parameters include the R multiplier (k), reporting rate (ρ), overdispersion parameter (ϕ), R_t scalar (c), and the location (ψ), shape (α), and scale (ν) parameters for the skew normal distribution. Blue indicates posterior estimates were higher than prior estimates, red indicates posterior estimates were lower than prior estimates, and grey indicates no difference. Areas in white indicate the corresponding model does not use that parameter. To calculate the relative difference, we subtracted the prior estimates of parameters from the posterior estimates of parameters and divided the difference by the prior to standardize over different parameter magnitudes. For comparison purposes, we left out the initial timing and initial location of ZIKV introduction parameters.

Reviewer #2 (Remarks to the Author):

The manuscript by Dr. Oidtman and colleagues describes an ensemble-modeling framework for the forecast of epidemiological dynamics associated with an emerging infectious disease. Specific reference is made to the Zika epidemic that hit Colombia in 2015-2016. The authors use a set of models whose features span three main axes: spatial structure (coupled vs. uncoupled, in the former case according to various hypotheses), temporal variability of the force of infection (dynamics vs. constant), and number of introduction events (one vs. two). Individual models are calibrated against surveillance data using a particle-filtering algorithm, which allows progressive assimilation of epidemiological information during the course of an outbreak in the form of updated estimates of the model parameters. In addition to the individual models' projections, also evaluated are the performances of ensembles obtained with either equal or suitably optimized weights. Performance is assessed in terms of the ability of each model (individual or ensemble) to forecast in space and time disease incidence or specific quantities of epidemiological interest. The analysis of different families of models and ensembles thereof allows the identification of possible trade-offs between the forecasting power of certain individuals models for specific epidemiological patterns vs. the overall potential of ensemble models averaged over the whole course of the epidemic. The take-home message is that ensembles of models accounting for different sources of uncertainty may be key to robust forecasts in the context of emerging infectious diseases. Although not completely a new one, this undoubtedly is an important message---also one that does ring a bell at a point in time when we all are facing the daily uncertainties associated with a pandemic caused by an emerging pathogen. The manuscript is quite well written and suitably organized; it is also supported by an extensive set of supplementary results.

All this being said, I have some technical comments that I would like to see addressed by the authors:

Comment 2.1

- The basic structure of the models is not completely clear, at least to me. From equation (4), one gathers that $I_{d,t}$ depends on $I''_{d,t}$, which, from equation (2), is defined as a function of $I'_{d,t-j}$ ($j=1...5$), which in turn is defined in equation (1) as a function on $I_{d,t}$. The lack of an epidemiological description of the underlying process does not help to solve this seeming circularity in the definition of disease incidence. I would suggest clarifying this methodological aspect that is common to all models.

Response 2.1

We adjusted the description of the model in response to this comment. Specifically, we clarified the distinction between the $I_{d,t}$ term (infections in the current time step) in Eq. 1 and the $I_{d,t+1}$ term (infections in the next time step) in Eq. 3-4, which was previously incorrectly specified.

We also added an introductory paragraph to the "Summary of models" section as a justification for why we chose to use this model:

To produce weekly forecasts of ZIKV transmission across Colombia, we sought to use a computationally efficient model with the flexibility to include relevant epidemiological and ecological mechanisms. We used a previously described semi-mechanistic, discrete-time, and stochastic model [55] that had been previously adapted and used to model mosquito-borne pathogen transmission [56, 57]. Using this model, we were able to account for the extended generation interval of ZIKV using overlapping pathogen generations across up to five weeks of the generation interval distribution of ZIKV [57]. Furthermore, we could specify this model to be either spatially connected or non-spatial—a key assumption that we considered in our analysis.

Comment 2.2

Relatedly, why did $I''_{d,t}$ also correspond to the dispersion parameter of the NB distribution in equation (4)?

Response 2.2

We added a more complete description and rationale for our model choice preceding the mathematical description of the model. Here, we explain that we are using an adapted version of a model previously described by Xia et al. (2004) (<https://www.journals.uchicago.edu/doi/10.1086/422341>), wherein they mathematically describe and justify this negative binomial derivation. In short, if we start with one infected individual and a per capita infection growth rate of β_t , then the number of infected individuals one generation later will be $\text{NegBin}(\beta_t, 1)$; given that we would start with I infected individuals instead of 1, then we would (approximately) have a sum of I negative binomials. $I_{t+1} \sim \text{NegBin}(\beta_t I_t, I_t)$ (the generalized version of our model) follows from the fact that the sum of b $\text{NegBin}(a, 1)$ distributions is $\text{NegBin}(a \times b, b)$ (Xia et al., 2004).

Comment 2.3

- Concerning human mobility, I understand that there might be some issues related to confidentiality, but the description of how mobile phone traces were used to inform spatial connectivity is quite obscure. For instance, what kind of information was stored in the CDRs? How was matrix H estimated? From line 246, it seems that the strength of pairwise spatial connections was assumed to be proportional to the number of calls between individuals living in (or calling from?) two (different) spatial units: is it so, and what is the rationale? Was it necessary to run a home-detection algorithm or was this piece of information readily available in the CDRs? In general, a more detailed (yet, obviously, fully privacy- and confidentiality-compliant) description of the CDRs and their use seems to be warranted, here.

Response 2.3

This is a very pertinent point. We updated the Data section of the methods to include a more complete description of the CDR data:

For models that relied on cell phone data to describe human mobility, we used anonymized and aggregated call detail records (CDRs). Every time a user receives or makes a call, a CDR including the time, date, ID, and the tower (BTS) providing the service is generated. The positions of the BTSs are georeferenced and so the aggregated mobility between towers can be tracked in time. We used this information to derive daily mobility matrices at the municipality level in Colombia from February 2015 to August 2015. Mobility matrices captured the number of individuals that moved in each given day from one municipality to another (i.e., that appeared in BTSs of different municipalities). The change for each day was captured by comparing the last known municipality to the current one. No individual information or records were available.

Comment 2.4

- As for the environmental drivers of transmission, I wonder whether some form of “hybridization” between the dynamic and static approaches has been attempted. The question arises because the static formulation includes, in addition to time-averaged proxies of temperature and vector abundance, a socioeconomic index that does not appear in the dynamic formulation. What is the reason for this difference? From the Methods section in the main text, I imagined it came from previous modeling efforts; from the Supplementary Information, I gathered that different “building bricks” were put together by the authors, resulting in the two different formulations. I believe that the reader should be given some more background information to fully appreciate the modeling choices made by the authors in this respect.

Response 2.4

This interpretation is correct. We sought to make this forecasting exercise as similar to if it were done in real time as possible. In this regard, we used results from two prior studies that modeled the relationship between ZIKV transmission and environmental drivers. In both of these studies, the data used to model this relationship was available *prior* to the beginning of the epidemic.

However, in an effort to make the dynamic and static models more comparable, we added a time-varying vector abundance component to the dynamic model, which originally only accounted for changes in temperature. Without this addition, we felt that the dynamic model may have been at a disadvantage to the static model and thus comparisons between these models would have been less straightforward. We

did not include the socioeconomic index, however, as it was not time-varying. We included this as a possible extension in the discussion on lines 285-287:

Relatedly, the static and dynamic R had minor differences in their formulations, such that the static R also included a socioeconomic index. In future work, it could be interesting to explore if the inclusion of this time-independent variable affected the dynamic R.

For further transparency for our model decisions, we added more background information in the beginning of section 4.2.2 to describe that this work was built on prior work and therefore included some different components on lines 420-422:

Specifically, both of these alternative formulations used different outputs from previous modeling efforts [6, 12] and because of this they contain slightly different components.

We additionally included a statement after we introduced the socioeconomic index to note that the PPP was not included in the dynamic model on line 444-445:

PPP_d is purchasing power parity in department d (a feature not included in the dynamic model) [60].

Comment 2.5

- The different model structures are characterized by different sets of parameters in need of calibration, each characterized by a specific cardinality. I wonder whether this has been (or can be) in any way used in the definition of the weights in ensemble forecasts.

Response 2.5

We appreciate this insight. To be candid though, we are not sure that we fully understand what is being suggested. Based on the way that our data assimilation algorithm works, similarities in the parameter sets across the different models are not utilized. While ignoring these similarities may be a missed opportunity with respect to model weighting somehow, the upside of this approach is that it ensures that the algorithm is generalizable to a wide range of models, regardless of what similarities they might have in terms of their parameters.

Alternatively, we could interpret this comment as suggesting that we weight the performance of models by the number of parameters and model fit to ensure we don't overfit the data with a model using many parameters. We could use AIC or DIC weights in this way. This is a fair suggestion, which we did consider doing. We ultimately did not perform model comparison in this way as we thought it could unfairly penalize the dynamic *R* models, which had three additional parameters to be "fitted," (albeit with very tight prior distributions to constrain deviation away from the established temperature-*R* relationship) simply because of the way we included the dynamic *R* relationship in our models.

We apologize if this response does not address the point that the reviewer is raising. We suspect that this may not be a critical issue with the manuscript and instead a possibly interesting direction for future work.

Comment 2.6

- I am a bit skeptical about producing forecasts of any sort before performing data assimilation (e.g. Figure 2). What is the merit in that, besides of course setting a baseline for the progressively increasing ability of the models to anticipate the epidemiological dynamics? Evidently, the choice of the initial prior distributions is not without consequences from a quantitative point of view, as the authors show that up to 20% of the originally sampled particles (hence, before the first data assimilation step) may remain until the end of the simulation period.

Response 2.6

For each particle, we produce a single forecast to "initialize" the model prior to data assimilation. There are two reasons we have to produce these forecasts prior to assimilation data. First, our process model is simulating infections, while the data we are assimilating into the model is reported cases (a fraction of all

infections). We assume that a significant portion of infections are not reported (a function of the reporting rate), including before cases were even reported. Consequently, to match low numbers of reported cases early in the epidemic, it is necessary to have higher levels of infection, which require time to build up. Second, two of the model parameters we are updating are only relevant for the pre-data reporting period: timing and location of the first infection(s). We assume the first infection occurs prior to the first reported case and therefore requires that we produce forecasts prior to case reporting.

Beyond the technical reasons why we initialized the model before data reporting began, ultimately our intuition was that a country could in theory produce a model and make forecasts from it in response to an emerging epidemic in another nearby region. More broadly, models can be and sometimes are used to make projections in the absence of data at the onset of an epidemic. Coincidentally, the "static R" formulation that we use here originated in a study that did precisely this for the Zika epidemic (Ref. 6), projecting its final size throughout the Americas without the use of any data from that epidemic. By comparing those projections to a retrospective estimate of the Zika epidemic's final size performed later (<https://journals.plos.org/plosntds/article?id=10.1371/journal.pntd.0008640>), it was found that those initial projections were actually quite reasonable.

We added a statement to the "Data assimilation and forecasting" section on line 463-464 to clarify why we had to start the models before any data reporting began:

For each particle, we produced a single forecast to "initialize" the model prior to the first reported case in Colombia.

Comment 2.7

- I would have expected a more detailed analysis of the performances of the equally- vs. optimally-weighted models. The authors state (line 190) that they generally found better performances associated with the former, but little quantitative evidence seems to be reported in the main text to support their observation. In fact, the optimally-weighted model is only used in Figure 3, where its overall forecast abilities cannot even be fully appreciated. By contrast, Figures 2, 4, and 5 only report results for the single models and the equally weighted ensemble. I think this comparison deserves a more quantitative assessment, also given its implications in the context of emerging pathogens, for which a "simpler" (and perhaps less data-greedy) equally-weighted approach sounds clearly appealing.

Response 2.7

We see this comment as being related to Comment 1.1 by Reviewer 1. In hindsight, this makes us realize that the message of our manuscript could have been presented more effectively. Although we do compare the equally and optimally weighted forecasts somewhat in the Discussion, the goal of the manuscript was to assess (in multiple ways) the forecast accuracy of specific models relative to an ensemble, rather than to compare ensembles.

Even though we did not view the comparison of these forecasts as a major focus of this manuscript, we have now done more to quantitatively compare ensemble forecasts in response to this comment. Specifically, in the figure wherein we explicitly compared the equally and optimally weighted forecasts (Fig. S23), we have now calculated the root mean square error (RMSE) as a quantitative assessment of the two forecasts and provided those results in the caption:

Observed versus forecasted incidence at 1-, 2-, 3-, and 4-week ahead intervals for EM-weighted and equally weighted ensemble models for Antioquia, Norte de Santander, Cauca, and Amazonas. Plotted departments reflect differences in population, epidemic size, and geographic regions of Colombia and are denoted by point type. Point shape denotes department. Point color indicates time at which the forecast was made (visually denoted in inset plot and color bar). Point is the median value and lines are the 50% credible interval. 1:1 line is in grey. The root mean square errors for the EM-weighted and equally-weighted forecasts shown here are 0.705 and 0.640, respectively.

In this way, we now demonstrate that the equally weighted ensemble provided slightly more accurate forecasts relative to the optimally weighted ensemble. We additionally now bring this up in the beginning of the discussion on equally weighted versus optimally weighted ensembles on lines 253-256 to further provide emphasis:

We considered both equally and optimally weighted ensembles and found that the equally weighted ensemble had a lower root mean square error than the optimally weighted ensemble (RMSE=0.640 and 0.705, respectively)—therefore providing slightly more accurate forecasts of the observed data (Fig. S23).

Minor comments

Comment 2.8

- line 90: why “retrospective”? This term and “real-time” seem to be used in opposition to describe target-oriented vs. incidence forecasts. However, the use of the two terms does not appear fully consistent. I would suggest sticking to consistent wording to help the reader navigate the different numerical exercises

Response 2.8

We removed any reference that our work was “real-time.” We now consistently use the terms target-oriented versus model-specific forecasts.

Comment 2.9

- Figure 4: those % changes are very small: can you comment on that?

Response 2.9

We added a sentence on line 163-165 to comment on this:

Such small changes in forecast performance when averaging over space shows that differences in forecast performance across space dominate relative to those across time.

Comment 2.10

- line 322: in the Methods section, which I read before the Results, the term “particle” is still undefined (of course the context becomes clear around line 330). I would suggest replacing it here with something different

Response 2.10

See Response 1.7, where we addressed the same issue in response to Comment 1.7 by Reviewer 1 (included below):

This is an important clarification that we were not previously making. We added a definition on line 77-81 (when we first introduce particles) to clarify what we mean by “particle”:

We observed a more substantial change in the proportion of individual stochastic realizations (where the n^{th} stochastic realization is the n^{th} “particle” generated from some set of parameters $\vec{\theta}_{t,n}$ at time t) resulting in an epidemic, with those particles resulting in no epidemic being filtered out almost entirely by week 12.

Throughout the “Data assimilation and forecasting” section, we made significant updates to the terminology to help with this clarification. We specifically denoted the difference between a parameter set ($\vec{\theta}_{t,n}$) and a particle ($\{I_{d,t,n}, C_{d,t,n}\}$), where a particle is the n^{th} set of state variables associated with the n^{th} parameter set. Throughout, we stopped using the term “particle” as loosely as we did previously.

Comment 2.11

- line 366: I know this will sound nitpicky, but $7,680 = 16 * 366 * 15 * (31 + 1)$, i.e. to get to the indicated total you need to include also the log scores at the national levels

Response 2.11

Thank you for pointing this out, we updated text to reflect the national levels too:

We evaluated a total of 7,680 log scores, reflecting three targets for each of 16 models in each of 31 departments plus at the national level and at each of 15 time points at which data assimilation occurred.

REVIEWERS' COMMENTS

Reviewer #1 (Remarks to the Author):

With my apologies to the authors for the delay in completing this review. I am satisfied with the changes that the authors have made and am happy to recommend this manuscript for publication.

Reviewer #2 (Remarks to the Author):

I have now read the revised manuscript by Dr. Oidtman and colleagues. It is my opinion that the authors have done a fair job of addressing the reviewers' comments and updating their manuscript accordingly. (I'm just a bit confused as to why many expressions have been erased from the manuscript just to be re-introduced verbatim afterwards, as shown by the diff version of the manuscript file, but that's a minor detail.) Specifically, I think that the presentation of many methodological aspects, in particular concerning the underlying transmission model, has remarkably improved. Similarly, the overall objectives of the paper are more clearly laid out, which certainly helps the reader follow the modeling choices made by the authors.

Now that model-related details are more plainly described, though, I see one more small point that might be worth a comment. Specifically, in the spatially connected models, the authors assume that between-department transmission can occur because of the contact between (i) mobile infectious and local susceptible people, or (ii) local infectious and mobile susceptible people (blue paragraph around line 393 of the diff version of the revised manuscript). In this description, the missing piece is transmission occurring because of contacts between mobile infectious and mobile susceptible people, possibly occurring in a department where both susceptible and infectious people happen to be traveling.

A practical example may perhaps help here. Let us consider New York City. Many residents of, say, Queens commute to Manhattan on a daily basis for work or study. There, they will find not only people who are resident in that borough, but also people who live elsewhere, say in Brooklyn, and who are also commuting for study or work to Manhattan. As a result, the total contacts between the residents of Queens and Brooklyn include not only those occurring because of the direct mobility fluxes between these two boroughs, but also those resulting from mobility to another borough (Manhattan, in this example) that attracts commuting fluxes from both of them. In epidemiological terms, a susceptible [an infectious] resident of Brooklyn commuting to Manhattan may be exposed to [contribute to onward transmission of] the pathogen because of social mixing with infectious [susceptible] people who either reside in Manhattan or are traveling there from another borough. This kind of mixing is seldom accounted for in spatially explicit epidemiological models, although the "Lagrangian" approach used by the authors should make it feasible to consider. The authors may thus want to add a comment on this point in the final version of the manuscript.

Finally, I see that the authors found one comments of mine (concerning differences in the cardinalities of the sets of calibration parameters for the different models) quite obscure, and I have to retrospectively agree---apologies for that. However, I was indeed thinking about some form of penalization to discourage overfitting (Akaike-like), and the authors proved enough prescient to formulate an answer for that.

Reviewer #1 (Remarks to the Author):

Comment 1.1

With my apologies to the authors for the delay in completing this review. I am satisfied with the changes that the authors have made and am happy to recommend this manuscript for publication.

Response 1.1

Thank you.

Reviewer #2 (Remarks to the Author):

Comment 1.1

I have now read the revised manuscript by Dr. Oidtmann and colleagues. It is my opinion that the authors have done a fair job of addressing the reviewers' comments and updating their manuscript accordingly. (I'm just a bit confused as to why many expressions have been erased from the manuscript just to be re-introduced verbatim afterwards, as shown by the diff version of the manuscript file, but that's a minor detail.) Specifically, I think that the presentation of many methodological aspects, in particular concerning the underlying transmission model, has remarkably improved. Similarly, the overall objectives of the paper are more clearly laid out, which certainly helps the reader follow the modeling choices made by the authors.

Response 1.1

We apologize for the mistake with the expressions. This error arose when producing the latexdiff file.

Comment 1.2

Now that model-related details are more plainly described, though, I see one more small point that might be worth a comment. Specifically, in the spatially connected models, the authors assume that between-department transmission can occur because of the contact between (i) mobile infectious and local susceptible people, or (ii) local infectious and mobile susceptible people (blue paragraph around line 393 of the diff version of the revised manuscript). In this description, the missing piece is transmission occurring because of contacts between mobile infectious and mobile susceptible people, possibly occurring in a department where both susceptible and infectious people happen to be traveling.

A practical example may perhaps help here. Let us consider New York City. Many residents of, say, Queens commute to Manhattan on a daily basis for work or study. There, they will find not only people who are resident in that borough, but also people who live elsewhere, say in Brooklyn, and who are also commuting for study or work to Manhattan. As a result, the total contacts between the residents of Queens and Brooklyn include not only those occurring because of the direct mobility fluxes between these two boroughs, but also those resulting from mobility to another borough (Manhattan, in this example) that attracts commuting fluxes from both of them. In epidemiological terms, a susceptible [an infectious] resident of Brooklyn commuting to Manhattan may be exposed to [contribute to onward transmission of] the pathogen because of social mixing with infectious [susceptible] people who either reside in Manhattan or are traveling there from another borough. This kind of mixing is seldom accounted for in spatially explicit epidemiological models, although the "Lagrangian" approach used by the authors should make it feasible to consider. The authors may thus want to add a comment on this point in the final version of the manuscript.

Response 1.2

This is a good point. We added a comment when we introduce the Lagrangian approach specifically noting that we account for transmission occurring in the first two manners you describe, but not in the third (highlighted in yellow).

By taking this Lagrangian approach to modeling human mobility, transmission across departments in our model can occur either by infected visitors transmitting to local susceptibles or susceptible visitors becoming infected by local infecteds, but not between infected visitors and susceptible visitors in a transient location. The relative occurrence of these events depends

on the prevalence of infection, susceptibility, local transmission potential, and mobility patterns of a given pair of departments.

Comment 1.3

Finally, I see that the authors found one comments of mine (concerning differences in the cardinalities of the sets of calibration parameters for the different models) quite obscure, and I have to retrospectively agree---apologies for that. However, I was indeed thinking about some form of penalization to discourage overfitting (Akaike-like), and the authors proved enough prescient to formulate an answer for that.

Response 1.3

Thank you.